# INFERENCE TIME LLM ALIGNMENT IN SINGLE AND MULTIDOMAIN PREFERENCE SPECTRUM

## ABSTRACT

Aligning Large Language Models (LLM) to address subjectivity and nuanced preference levels requires adequate flexibility and control, which can be a resource-intensive and time-consuming procedure. Existing training-time alignment methods require full re-training when a change is needed and inference-time ones typically require access to the reward model at each inference step. To address these limitations, we introduce inference-time model alignment method that learns encoded representations of preference dimensions, called *Alignment Vectors* (AV). These representations are computed by subtraction of the base model from the aligned model as in model editing enabling dynamically adjusting the model behavior during inference through simple linear operations. Even though the preference dimensions can span various granularity levels, here we focus on three gradual response levels across three specialized domains: medical, legal, and financial, exemplifying its practical potential. This new alignment paradigm introduces adjustable preference knobs during inference, allowing users to tailor their LLM outputs while reducing the inference cost by half compared to the prompt engineering approach. Additionally, we find that AVs are transferable across different fine-tuning stages of the same model, demonstrating their flexibility. AVs also facilitate multidomain, diverse preference alignment, making the process 12x faster than the retraining approach.

## 1 INTRODUCTION

Aligning LLMs is crucial for adapting them to meet human preferences. Standard training-time alignment methods, such as RLHF (Ouyang et al., 2022) and DPO (Rafailov et al., 2024), are conducted during model training. However, making nuanced preference adjustments during inference with these approaches would necessitate retraining, which requires substantial amounts of time, preference data and computational resources. Inference-time LLM alignment, by contrast, delays the alignment process until inference (Wang et al., 2024). While preference alignment can be achieved through training-time methods or targeted prompting, fine-grained control over preferences at inference remains largely unexplored in current State-of-the-Art (SOTA) works (Sahoo et al., 2024; Guo et al., 2024). This research introduces an inference-time model editing technique via *Alignment Vectors* (AV), offering users dynamic preference adjustments without additional computational overhead.

Due to their extensive capabilities, LLMs are now employed in different fields, including the ones that requires specialized domain understanding like legal (Guha et al., 2024), healthcare (Yang et al., 2023) and financial (Huang et al., 2023) industry. However, the diverse needs of a broad customer base require that LLM outputs be carefully refined. For instance, while a healthcare provider might need detailed medical responses for professional use, a public health forum may prefer more generalized information to avoid misinterpretation. Although prompt engineering can temporarily address these needs, it becomes costly when scaled (Li et al., 2023).

Furthermore, managing multiple alignment objectives can be complex. Consider an insurance company that needs expert legal responses, generic financial answers, and to avoid medical responses; balancing these demands poses a significant challenge. A joint training with targeted preference levels can resolve the problem, however, it lacks flexibility, and training can be resource intensive. Hence, at present, there is no work that addresses such preference flexibility in the inference time.

Thus, developing flexible, inference-time adjustable model alignment to manage costs and maintain efficiency in the long term remains a major research gap.

In the current literature, preference dimensions such as helpfulness, harmlessness, and honesty are well-studied (Bai et al., 2022; Ji et al., 2024). Some studies also explore the controllability of these dimensions by numerically categorizing preference "levels" (Guo et al., 2024). However, specialized dimensions have a finer granularity which gives more control in making adjustments during inference time. Hence, to simplify the controllability objective, we primarily focus on achieving meaningful preference tunability by emphasizing proficiency levels in responses within specialized domains. Additionally, we demonstrate preference tunability in a general domain, specifically 'safety' in this case.

It is important to note that existing literature lacks specialized preference alignment datasets for domain-specific Query-Response pairs. To fill this gap, we propose a method to generate both queries and responses synthetically. Our queries are derived from personas sampled from the PersonaHub dataset (Chan et al., 2024) and supplemented by novel personas created through prompts to an LLM. We produce responses at three levels: avoidance (Avd), generic response (Gen), and expert opinion (Exp).

In addition, to achieve inference time preference tunability, we propose a simple model editing technique called Alignment Vector Arithmetic, which is based on the concept of *Task Arithmetic* (Ilharco et al., 2023). AVs can be obtained by directly subtracting the base model parameters from the aligned model, and can be added in the inference time. Hence, our first research question (**RQ1**) Are alignment vectors valid representation of the preference dimensions? To address this question, we systematically integrate the alignment vector into the base model with varying weights, both positive and negative, and analyze the resulting changes in model behavior. Our second research question is posed as (**RQ2**) Can we calibrate different alignment vectors to achieve diverse multi-domain preference? We address RQ2 through different domain-specific AV-integration strategy.

The key contribution of this work are:

- We frame LLM alignment in single and multiple domains as a model editing problem and introduce an inference-time tunable mechanism, which allows flexible adjustment of generation output along the preference dimension.
- We generate a synthetic dataset with a total of 38k queries, each paired with responses categorized into three levels of specialized subject matter proficiency across three specialized domains: Medical, Financial, and Legal. The dataset will be available through this link.
- By adjusting the merging coefficients, we achieve diverse, multidomain behaviors efficiently, saving time and resources. Unlike joint training, which requires $p^D$ adjustments for $D$ domains and $p$ preference levels, our method only requires $D$ training runs, reducing resource usage by a factor of $p^D/D$.

> **Why inference time alignment over conventional approach?** In contrast to conventional training time approaches, inference time alignment provides flexibility and adaptability by enabling dynamic adjustments to model behavior based on task or user needs without retraining.

## 2 RELATED WORKS

Research on inference time alignment has explored several approaches, with prompt engineering being the simplest and most basic. Techniques, such as zero-shot, few-shot, and Chain-of-Thought (COT) prompting have proven effective in aligning language model responses to user queries during inference time (Radford et al., 2019; Sahoo et al., 2024; Wei et al., 2022). However, prompt engineering comes with expensive inference time and cost and that could be infeasible when scaled. Additionally, effective prompt engineering assumes that the user is skilled at interacting with LLMs (Meskó, 2023; Oppenlaender et al., 2023). In contrast, our approach does not rely on prompting, and thus meets the diverse needs of users.

Li et al. introduced Inference-Time Intervention (ITI), which identifies a sparse set of attention heads with high linear probing accuracy for a target task and shifts their activation along task-correlated

directions during inference time (Li et al., 2024). However, their methods are largely applicable to truthfulness and not controllable. A related approach involves learning Safety Related Vectors (SRV), to steer harmful model outputs towards safer alternatives (Wang et al., 2024). However, the lack of controllability and input dependency of this technique to determine if the prompt may induce unsafe response limits its applicability in our context. Huang et al. introduced DeAl, an alignment method that treats alignment as a heuristic-guided search process (Huang et al., 2024). However, this approach significantly slows down the decoding process due to the expansion of the search space. Liu et al. studied regularization strength between aligned and unaligned models to have control over generation (Liu et al., 2024). Although closely related to our work, their method lacks clarity on whether fine-grained preference levels can be achieved, and it appears less flexible in transferring alignment behavior across different contexts. Researhers controlled attributes of generated contents by adding control token in the prompt (Guo et al., 2024; Dong et al., 2023). Despite its effectiveness, this method requires training LLMs with a particular data format, which restricts the flexibility of control during inference.

Rame et al.'s work is closely related to our multi-domain preference alignment (Rame et al., 2024). However, their approach focuses on training-time alignment by interpolating weights from models fine-tuned on diverse rewards to achieve Pareto-optimality. In contrast, our work introduces a preference adjustment strategy that operates at inference time, in addition to achieving multi-dimensional alignment. Similarly, while Jang et al. address personalized preference alignment and post-hoc merging, our approach provides a unique capability: preference level adjustment Jang et al. (2023). Our research addresses the existing gaps by exploring model editing techniques to achieve fine-grained control over preferences during inference, without the constraints of prompt engineering or additional training requirements.

## 3 METHODOLOGY

### 3.1 OBTAINING ALIGNMENT VECTOR

To obtain the *Alignment Vector* (AV), we first perform alignment through DPO algorithm, using an 'ipo' loss function to create a domain-specific aligned model (Rafailov et al., 2024; Azar et al., 2024). We derive AV using a direct subtraction technique. This method, inspired by the work of Ilharco et al., involves performing task arithmetic to capture alignment vectors (Ilharco et al., 2023). Their approach suggests that by subtracting the base pre-trained model parameters from those of a fine-tuned model (specifically fine-tuned on a given task), a task-specific direction is encoded. Moving the model along this task direction enhances its performance on that particular task.

We build AV by subtracting the weights of an unaligned model from the weights of the same model after alignment fine-tuning on a task. If $\theta_{aligned}$ denotes the model parameter after fine-tuning on a preference dimension, then the Alignment Vector can be obtained by the following:

$$\theta_{AV} = \theta_{aligned} - \theta_{unaligned} \tag{1}$$

### 3.2 SINGLE DOMAIN ALIGNMENT

To enable preference tunability across different domains, we perform a weighted integration of the alignment vectors into the base (or unaligned) model, where the weights can be both positive and negative. Our hypothesis is that this gradual integration will result in a corresponding gradual increase or decrease in the model's proficiency. This process is governed by the following equation:

$$\theta_{aligned} = \theta_{unaligned} + \lambda * \theta_{AV} \tag{2}$$

By adjusting the value of $\lambda$, we aim to control the proficiency of the model's generated responses. Assuming when $\lambda = 0$, the model remains unaltered and functions as the base, unaligned model. If the $\theta_{AV}$ encodes the expert behavior in a certain domain, as $\lambda$ increases towards 1, the model becomes increasingly aligned, achieving full proficiency at $\lambda = 1$.

We further hypothesize that when $\lambda$ takes on negative values, the model's behavior tends to reverse the preference ranking. For instance, if the base model typically generates generic responses and the aligned model is designed for expert-level responses, moving $\lambda$ in the negative direction will

shift the model towards avoidance behavior. Therefore, to control the proficiency of the responses, adjusting $\lambda$ is sufficient, eliminating the need to train the model with a new preference configuration.

### 3.3 MULTIDOMAIN ALIGNMENT

When dealing with multiple domains simultaneously, the interaction between these domains can present a significant challenge. While each individual preference vector encodes domain-specific attributes, they also embed proficiency levels which can easily generalize and negatively affect multidomain diverse behavior. This complexity can make it difficult to integrate multiple domains effectively.

Our goal is to achieve a diverse multidomain preference, which we approach by using the following equation:

$$\theta_{multidom\_aligned} = \alpha * \theta_{AV\_dom1} + \beta * \theta_{AV\_dom2} + \gamma * \theta_{AV\_dom3} \tag{3}$$

In this equation, $\alpha$, $\beta$ and $\gamma$ represent the integration coefficients for the domains in question, respectively. By identifying different sets of these coefficients, we aim to achieve varying levels of preference across the three domains.

## 4 SYNTHESIZING SPECIALIZED PREFERENCE DATA

To gather data for preference tuning on response proficiency levels, we employ two methods to collect queries: "PersonaHub" (Chan et al., 2024) and "CreatePersona." We also prompt an LLM to generate responses at three distinct proficiency levels. The quality of these responses is then assessed through human evaluation. Figure 1 provides a detailed overview of the entire data collection process.

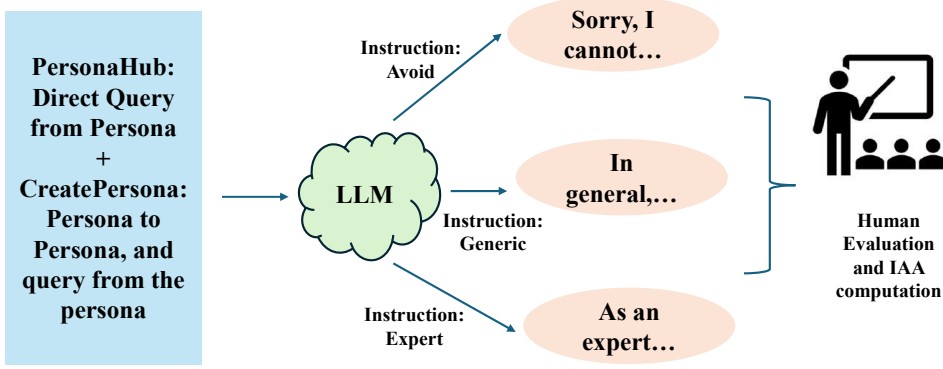

Figure 1: The process of data collection. Personas are sourced from both the PersonaHub dataset and the CreatePersona method. These personas are then fed to an LLM to generate queries. The LLM is prompted with specific instructions to produce responses across three proficiency levels. Following this, human evaluation is conducted to ensure the accuracy and quality of the generated response levels.

### 4.1 QUERY GENERATION

First, we randomly select 7,000 personas from the PersonaHub dataset, which contains 200,000 personas (Chan et al., 2024). Using these selected personas, we prompt an LLM, Claude-3-Sonnet (Anthropic, 2024), to generate specialized domain queries. We chose to use Claude-3-Sonnet over GPT-4 for two main reasons: First, Claude-3-Sonnet has consistently demonstrated performance on par with GPT-4, often ranking among the best foundational models. Second, we opted to use GPT-4 as an independent evaluator and sought to mitigate the known bias where evaluators tend to favor their own outputs over those generated by other models (Zheng et al., 2024; Anthropic, 2024).

To enhance the diversity of our dataset and create a more robust hub, we then initiate a hierarchical generation process called "CreatePersona." We begin by randomly generating a few persona-query pairs using Claude. Our initial investigations revealed that generating too many initial pairs often reduces diversity. Therefore, we limit this to five initial persona-query pairs. From each initial persona, we recursively generate additional persona-query pairs that are relevant to the root persona. We randomize this process three times. After a thorough clean-up, involving truncation, and reformatting, we obtained 13,000 personas for the medical domain, 12,374 personas for the financial domain, and 12,867 personas for the legal domain. Each persona is accompanied by queries pertinent to their respective specialized domains.

## 4.2 RESPONSE GENERATION

We generate the response from the queries into three distinct levels: avoidance of response (Avd), generic response (Gen), and expert response (Exp). Detailed instructions are provided to the LLM to facilitate the generation of these responses (see Appendix C). Furthermore, we observe a progressive increase in response length from the avoidance level to the expert level. To mitigate potential bias associated with response length, we instructed the LLM to produce responses of random lengths.

## 4.3 HUMAN EVALUATION OF MULTI-LEVEL RESPONSE GENERATION

To evaluate the quality of the generated responses, we conduct a small experiment involving three annotators. Each annotator is asked to categorize a set of LLM-generated responses into one of three categories: Avd, Gen, and Exp. We provide the annotators with clear definitions of these categories. Each annotator reviews 30 queries along with their three-level responses, with at least 15 examples shared between every pair of annotators. This allows us to compute the average Cohen's kappa score, which is found to be 0.84 (Cohen, 1960), indicating substantial agreement among the annotators.

We also calculate the average annotation agreement for each annotator with the LLM generation. Responses generated with the Avoidance instruction have the fewest disagreements or misclassifications. However, some Gen and Exp responses are occasionally misclassified from one another. We observe that certain responses, although aligned with the expert spectrum, are misidentified as generic due to their tone, and vice versa. Additionally, a few avoidance responses provide basic information, leading to their misclassification as Gen responses. These findings suggest that the levels may represent a continuous spectrum rather than distinct categories, highlighting the need for further research to more precisely define these proficiency levels.

## 5 EXPERIMENTS

### 5.1 EVALUATION METRIC

To assess the performance after alignment, we use a metric called *preference accuracy* (pref. acc). This metric reports the accuracy at each alignment level. To calculate it, we first compute the token-level mean log-probability for each of the three response levels across all queries for the aligned model. Then, for each sample in the validation set, we determine which alignment level has the highest log-probability. For example, in proficiency level alignment, it can be among Exp, Gen, and Avd. Finally, we report the percentage of samples where each alignment level had the highest log-probability in the validation set. A higher preference accuracy in an alignment spectrum indicate the dominant behavior of that level.

To illustrate, for a query $q \in Q$, the mean log-probability for response $r \in R$, where $R$ can be different alignment levels, is computed for model $M_\lambda$ as:

$$MeanLogProb(r, q, M_\lambda) = \frac{1}{T_r(q)} \sum_{i=1}^{T_r(q)} \log P(t_i \mid \text{context}, M_\lambda),$$

where $T_r(q)$ is the response length. The preferred alignment level is:

$$r^*(q) = \arg\max_{r \in R} MeanLogProb(r, q, M_\lambda).$$

The preference accuracy for level $r$ is:

$$Pref.\ Acc(r) = \frac{1}{|Q|} \sum_{q \in Q} \mathbf{1}[r^*(q) = r],$$

where $\mathbf{1}[r^*(q) = r]$ is the indicator function. Higher *Pref. Acc(r)* indicates the dominant behavior of the preference alignment level $r$. A similar approach was also used in pairwise preference accuracy computation in (Stiennon et al., 2020).

Additionally, we use an auxiliary metric as "GPT-4 judged generation accuracy", where we generate the responses from queries in a sample, and ask GPT-4 to annotate it as one of the three levels (Zheng et al., 2024). After that, we simply report the percentage of each annotated alignment level.

## 5.2 BASELINE APPROACHES

Since no existing model-editing methods currently support inference-time controlled alignment, we use a 'prompting' approach as our baseline. This method involves manually instructing the LLM through prompts to generate responses at different proficiency levels based on predefined definitions. Notably, unlike model editing, the 'prompting' approach can help the model achieve discreet levels instead of spectrums.

Our second baseline, aimed at achieving multidomain diverse behavior, is the 'Joint Training' approach. In this method, we combine data from various domains to create a preference dataset tailored to different proficiency levels. Although this approach is applied during training rather than at inference time, it provides valuable insights for establishing diverse alignment objectives.

Additionally, we report the performance of the model when we simply prompt the query without providing any additional instruction or performing model editing. We refer to this as the 'default' performance.

## 5.3 MODEL AND TRAINING CONFIGURATION

While we define three primary preference levels for specialized domain proficiency, our approach can be extended to accommodate additional levels if needed. For DPO training, we employ a full fine-tuning strategy, using a fixed beta parameter of 0.1. During alignment training, we focus on tuning our model to the "expert" proficiency level within each domain, where "expert" is preferred over "generic," and "generic" is preferred over "avoidance." To show the preference tunability, we experiment with different $\lambda$ values, and we choose an interval of 0.1. We empirically found that an interval of 0.1 provides a fine-grained and practical resolution, allowing us to capture significant shifts in the model's behavior.

As a base model for alignment, we use *Mistral-7B-Instruct-v0.3* (Jiang et al., 2023). We conducted our training on NVIDIA A100 GPUs and we utilize 80% of the generated data in each domain for training and 20% for testing. For the validation process, we allocated 3% of the data for training time validation. We used a batch size of 4 and trained each model for one epoch, monitoring the validation loss to determine when to stop training.

## 6 RESULTS AND DISCUSSION

### 6.1 SINGLE DOMAIN PREFERENCE TUNING

In the context of single-domain inference-time preference tuning, we only use the AV derived by aligning the model to generate responses at an expert-level within a given domain. The primary objective of this tuning process is to facilitate model editing that introduces a tunable parameter, allowing the user to control the proficiency level of the generated responses in a continuum. Consequently, one alignment vector is established for each domain, enabling the model to navigate and produce output across varying spectra of proficiency. This, in turn, also addresses **RQ1**.

From Table 1, 2, and 3 we observe that the baseline of adding an instruction to demonstrate a specific expertise spectrum does not significantly improve preference accuracy. Additionally, these

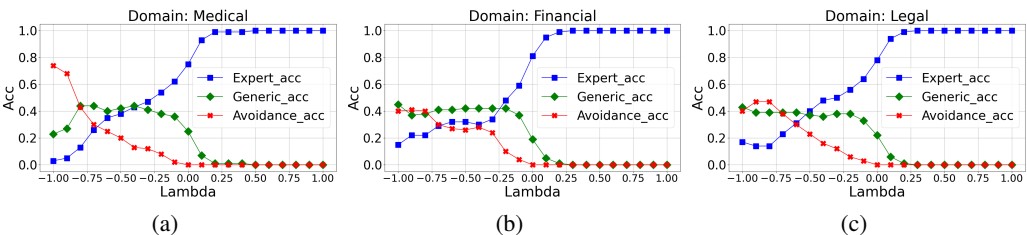

(a)           (b)           (c)

Figure 2: By changing the $\lambda$ parameter in the model editing process, we achieve different alignment objectives. In (a), when $\lambda > 0.3$, we find the model aligning with expert answers to medical queries by prefering expert responses over the others. However, when $\lambda < -0.8$, we see the model prefers avoidance of responses. In between these points, we observe the model answering generically to medical queries. (b) and (c) demonstrates this behavior for financial and legal domains respectively. Here $\lambda$ acts as a "tunable knob", through which users can adjust the behavior of the model, and have the expertise level at any spectrum they want

| | **Lambda** | **Dominant Alignment Behavior** | **Med pref. acc** | | | **GPT-4 judged gen. acc** | | |
|---|---|---|---|---|---|---|---|---|
| | | | **Exp** | **Gen** | **Avd** | **Exp** | **Gen** | **Avd** |
| Default | 0 | | .75 | .25 | 0 | .90 | .05 | .05 |
| Prompting | 0 | Exp | .78 | .22 | 0 | .90 | .05 | .05 |
| | 0 | Gen | .69 | .31 | 0 | .50 | .50 | 0 |
| | 0 | Avd | .60 | .25 | .15 | .15 | .55 | .30 |
| Ours: Model Editing | .5 | Exp | **.95** | 0 | .05 | **1.0** | 0 | 0 |
| | -0.7 | Gen | .26 | **.44** | .30 | 0 | **.60** | .40 |
| | -1.2 | Avd | .03 | .13 | **.84** | .05 | .20 | **.75** |

Table 1: How model editing performs to steer the Medical Domain Expertise response level. Lambda = 0 indicates the model with no alignment. Tuning Lambda to different values with our model editing approach leads to varying levels of proficiency responses. As such, we observe Exp, Gen, and Avd behavior just by aligning one model.

| | **Lambda** | **Dominant Alignment Behavior** | **Fin pref. acc** | | | **GPT-4 judged gen. acc** | | |
|---|---|---|---|---|---|---|---|---|
| | | | **Exp** | **Gen** | **Avd** | **Exp** | **Gen** | **Avd** |
| Default | 0 | | .81 | .19 | 0 | .85 | .15 | 0 |
| Prompting | 0 | Exp | .84 | .16 | 0 | .95 | .05 | 0 |
| | 0 | Gen | .57 | .43 | 0 | .75 | .25 | 0 |
| | 0 | Avd | .35 | .49 | .16 | .20 | .60 | .20 |
| Ours: Model Editing | .30 | Exp | **.85** | .15 | 0 | **1.0** | 0 | 0 |
| | -0.40 | Gen | .30 | **.42** | .28 | .35 | **.50** | .15 |
| | -1.4 | Avd | .07 | .20 | **.73** | 0 | .15 | **.85** |

Table 2: How model editing performs to steer the Financial Domain Expertise response level. Similar to the medical domain, we observe that the proficiency levels can be controlled in the inference-time only by varying Lambda.

instruction-augmented prompts are nearly as long as the original queries, which doubles the inference cost. Notably, both the training and validation datasets were curated using prompts from a different language model (Claude-3-Sonnet). Despite this, the base model (Mistral-Instruction) achieves high accuracy for expert-level answers, even without additional instructional prompts. The lower accuracy observed in the generic (0.31) and avoidance (0.15) categories with the prompt suggests the model's limited ability to accurately distinguish responses that align with the given instruction.

| | Lambda | Dominant Alignment Behavior | Leg pref. acc | | | GPT-4 judged gen. acc | | |
|---|---|---|---|---|---|---|---|---|
| | | | **Exp** | **Gen** | **Avd** | **Exp** | **Gen** | **Avd** |
| Default | 0 | | .78 | .22 | 0 | .85 | .15 | 0 |
| Prompting | 0 | Exp | .79 | .21 | 0 | 1.0 | 0 | 0 |
| | 0 | Gen | .59 | .41 | 0 | .65 | .35 | 0 |
| | 0 | Avd | .41 | .30 | .29 | .15 | .40 | .45 |
| Ours: Model Editing | .30 | Exp | **1.0** | 0 | 0 | **1.0** | 0 | 0 |
| | -0.70 | Gen | .23 | **.39** | .38 | 0 | **.65** | .35 |
| | -1.4 | Avd | 0 | .20 | **.80** | 0 | .05 | **.95** |

Table 3: How model editing performs to steer the Legal Domain Expertise response level. The pattern of inference-time flexibility continues for the legal domain as well. By tuning the model in one direction (legal expert), we can achieve other proficiency levels as well.

For our model editing approach, we add the AV with different proportions of $\lambda$ (Lambda). We observe that steering $\lambda$ in the negative direction decreases the likelihood of generating expert responses, with an avoidance behavior emerging at $\lambda = -1.2$. In the medical domain, the model displays generic behavior when $\lambda$ is set to -0.7 and achieves full expertise at $\lambda = 0.5$.

Figure 2 illustrates the tunable nature of the preference expertise spectrum across all three domains. Notably, at $\lambda = 0$, the model predominantly generates expert responses in all domains. In the medical domain, the model reaches the higher end of the expertise spectrum when $\lambda$ exceeds 0.3. Between $\lambda = -0.4$ and $\lambda = -0.8$, the model exhibits varying degrees of generic behavior and beyond that, the model starts behaving with topic avoidance.

Next, we investigate if the gradual model editing method also impacts the performance in the other domains. Our findings indicate that the specialized behavior is indeed reflected across various domains, even when the AV is extracted for a specific domain. For instance, Table 4(a) demonstrates that the addition of a medical AV with $\lambda = 0.5$ also enhances the model's expertise in the financial domain. Similarly, we observed that with $\lambda = -1.2$ the model exhibits avoidance behavior in both the legal and financial domains. This pattern is consistent when using other specialized domain vectors, such as financial and legal, as shown in Tables 4(b), 4(c).

**Effect on General Alignment** We also examine whether model editing for controllable proficiency levels influences the general domain preference (i.e., 'helpfulness' and 'safety'). Notably, we do not observe any regression in the safety domain; however, the model becomes increasingly helpful as $\lambda$ increases. With the rise in $\lambda$, the model provides more detailed and specific guidance, which aligns with human preferences for helpfulness. Conversely, decreasing $\lambda$ causes the model to avoid answering, which is perceived as unhelpful. Notably, the range of change in general domain preference accuracy is $\pm18\%$ for helpfulness and $\pm12\%$ for safety, indicating that model editing does not lead to significant regression in general domain performance.

## 6.2 Multi Domain Preference Tuning

In multi-domain preference tuning, we observe distinct behaviors across different domains by adjusting specific configurations. Since, we have three proficiency levels, accuracy higher than 33% and the highest among the three levels can be considered as the "dominant" proficiency level. For example, as shown in Table 5, we find that an AV-based editing coefficient of -1, -1, and 0.6 for the Medical, Financial, and Legal domains, respectively, results in *avoidance* being the dominant behavior in the Medical and Financial domains, with accuracies of 0.46 and 0.42, respectively, and *expertise* being dominant in the Legal domain, with an accuracy of 0.78. Therefore, we address **RQ2** as well.

It is important to note that there are 27 possible combinations (three domains, each with three behavioral spectrums), and through a grid search of model editing configurations, we found that the model can exhibit 22 combinations where the desired behavior is dominant in different domains. When compared with baseline joint training, the accuracy in joint training is near-perfect. Note that

| Lambda | Fin pref. Acc | | | Leg pref. Acc | | | General Pref. Acc | | | |
|---|---|---|---|---|---|---|---|---|---|---|
| | | | | | | | Safety | | Helpfulness | |
| | Exp | Gen | Avd | Exp | Gen | Avd | Safe | Unsafe | Helpful | Unhelpful |
| 0 | .81 | 19 | 0 | .78 | .22 | 0 | .58 | .42 | .60 | .40 |
| 0.5 | 1.0 | 0 | 0 | 1.0 | 0 | 0 | .58 | .42 | .66 | .34 |
| -0.7 | .59 | .40 | .01 | .58 | .32 | .10 | .57 | .43 | .58 | .42 |
| -1.2 | .03 | .20 | .77 | .08 | .18 | .74 | .57 | .43 | .49 | .51 |

(a) Out of Domain (special and general) preference accuracy for Medical domain responses.

| Lambda | Med pref. acc | | | Leg pref. acc | | | Gen pref. acc | | | |
|---|---|---|---|---|---|---|---|---|---|---|
| | | | | | | | Safety | | Helpfulness | |
| | Exp | Gen | Avd | Exp | Gen | Avd | Safe | Unsafe | Helpful | Unhelpful |
| 0 | .75 | .25 | 0 | .78 | .22 | 0 | .58 | .42 | .60 | .40 |
| .30 | .97 | .02 | .01 | .98 | .02 | 0 | .57 | .43 | .59 | .41 |
| -.40 | .61 | .37 | .02 | .57 | .35 | .08 | .59 | .41 | .57 | .43 |
| -1.4 | .18 | .40 | .42 | .19 | .52 | .29 | .55 | .45 | .51 | .49 |

(b) Out of Domain (special and general) preference accuracy for Financial domain responses

| Lambda | Med pref. acc | | | Fin pref. acc | | | Gen pref. acc | | | |
|---|---|---|---|---|---|---|---|---|---|---|
| | | | | | | | Safety | | Helpfulness | |
| | Exp | Gen | Avd | Exp | Gen | Avd | Safe | Unsafe | Helpful | Unhelpful |
| 0 | .75 | .25 | 0 | .81 | .19 | 0 | .58 | .42 | .60 | .40 |
| .30 | 1.0 | 0 | 0 | 1.0 | 0 | 0 | .53 | .47 | .59 | .41 |
| -.70 | .30 | .57 | .13 | .32 | .56 | .12 | .56 | .44 | .53 | .47 |
| -1.4 | .20 | .58 | .22 | .13 | .50 | .37 | .49 | .51 | .51 | .49 |

(c) Out of Domain (special and general) preference accuracy for Legal domain responses

Table 4: Observing the generalization effect of our model editing approach. Here, we gradually add the in-domain AV with the base model, and observe the performance for out-of-domain proficiency levels. We find that steering the proficiency levels in one domain also generalizes across other domains.

| Baseline: Joint training | | | Ours: Model Editing | | | editing coef |
|---|---|---|---|---|---|---|
| Med | Fin | Leg | Med | Fin | Leg | |
| Avd (100%) | Avd (99%) | Exp (98%) | Avd (46%) | Avd (42%) | Exp (78%) | [-1, -1, .6] |
| Avd (100%) | Exp (91%) | Exp (94%) | Avd (43%) | Exp (44%) | Exp (80%) | [-1, .8, .6] |
| Avd (100%) | Exp (90%) | Avd (90%) | Avd (57%) | Exp (56%) | Avd (36%) | [-.4, .4, -.8] |
| Exp (99%) | Avd (100%) | Exp (97%) | Exp (88%) | Avd (44%) | Exp (87%) | [.2, -.8, -.2] |

Table 5: Multidomain expertise can be achieved by model editing. In the baseline joint training approach, we find near-perfect performance, however, we need to perform separate training for each specific configuration. On the contrary, once trained on domain specific expertise, we can perform inference time adjustment and obtain specific configuration to behave in different way in each of the domain.

multi-domain expertise behavior can be achieved by training data for expertise behavior in each domain individually, requiring only three instances of DPO training. In contrast, joint training requires 27 separate training instances, demanding nine times more resources and time.

To compare the targeted training approach with our approach, each job, along with its corresponding validation runs, takes about 72 hours of training on A100 GPUs. This adds up to a total of 72 * 27 = 1,944 hours of training time. In contrast, the grid search method, which evaluates 21 coefficient values across three domains, results in 21 * 21 * 21 = 9,261 evaluation cycles. Since each evaluation takes around 60 seconds, the total time is approximately 155 hours—making it 12 times faster than

the full training approach. However, one can employ a hierarchical search approach, which can further reduce the search space, and thus, the resource usage.

However, unlike single-domain preference tuning, achieving continuous tunability across multiple domains presents significant challenge. Our observations suggest that single-domain model editing often leads to over-generalization, which, in turn, compromises the precision required for fine-tuning behaviors across multiple domains. This over-generalization effect may result from the model's inherent tendency to generalize learned behaviors beyond the specific domain for which they were tuned in the first place.

### 6.3 CAN AV BE EXTENSIBLE FOR GENERAL DOMAIN?

To explore the generalizability of model editing by AVs across various domains, we focus on the safety alignment aspect. We start by aligning our base model towards the "safety" dimension by obtaining the safety AV and gradually integrating it with the base model. For the safety alignment, we use the PKU-SafeRLHF dataset, and the sample the examples where chosen response is labeled safe, and the rejected response is labeled unsafe (Ji et al., 2024). We compute the pref. acc in the same way described in 5.1, where $R = \{safe, unsafe\}$.

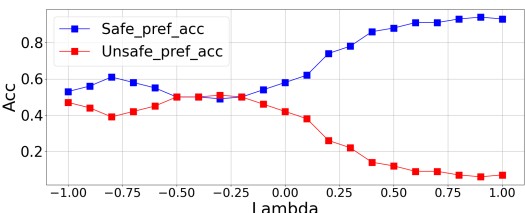

Figure 3: Controlling safety by model editing

Figure 3 illustrates that the model exhibits mixed safety accuracy initially when $\lambda = 0$ with a safety preference accuracy of 0.53 and an unsafe preference of 0.47. As $\lambda$ increases, the model progressively aligns more with safety, achieving a safety preference accuracy of 0.93 at $\lambda$=1. However, when $\lambda$ is adjusted negatively, the safety scores become inconsistent and mixed. Notably, even at large negative $\lambda$ values, the model does not become fully "unsafe".

In constructing the response proficiency levels, we intentionally maintain three distinct spectrums. In contrast, the PKU-SafeRLHF dataset does not follow this structure, as it lacks any specific gradation in safety levels. Moving forward, we plan to collect a dataset with gradual safety levels, which may improve controllability in general domains.

### 6.4 ANALYZING THE TRANSFERABILITY OF ALIGNMENT VECTOR

Next, we explore whether AVs derived from an instruction-tuned model can be effectively applied to a different model within the same family but at a different stage of fine-tuning. As a case study, we select a safety-aligned version of the base model, trained on the PKU-SafeRLHF safety preference dataset, to assess the transferability of these alignment vectors. Using a similar approach to single-domain model editing, we gradually integrate the AVs into our target model, which is safety-aligned.

Figure 4 presents the model's performance as $\lambda$ is varied. Our findings indicate that when $\lambda$ is adjusted from -1 to +1, the model's behavior related to safety—its primary control objective—remains relatively stable. For instance, in the medical domain (Figure 4(a)), varying $\lambda$ within this range results in a minimal change in safety preference accuracy, with a difference of only 0.11 between the lowest and highest accuracy points. In contrast, the accuracy of medical expert response preferences improves significantly, with an increase of 0.81—over seven times greater than the change in safety preference accuracy. Hence, we can conclude that, the AV obtained by our method is trasferable to models aligned on other orthogonally aligned objectives as well.

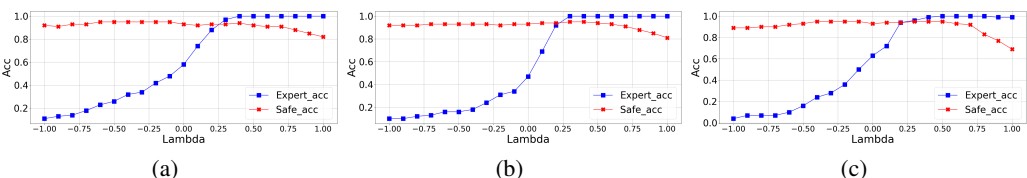

Figure 4: Effect of proficiency-level-encoded Alignment Vectors integration with a safety-aligned model. (a) Medical domain (b) Financial Domain (c) Legal Domain proficiency control

## 7 LIMITATIONS AND FUTURE WORK

Our work has several limitations and areas for future research. First, we used a basic approach for obtaining alignment vectors, but more advanced techniques like parameter thresholding, zeroing, or SVD-based separation should be explored (Yadav et al., 2024; Gao et al., 2024). Second, our method works only for LLMs with the same architecture, so applying it to different architectures is a potential direction for future study. Third, we tested our approach only on Mistral-7b, so validation with other open-source LLMs and SLMs is necessary. Fourth, we did not evaluate the correctness of the specialized domain responses, which we plan to do in the next version of our dataset. Lastly, we relied on an extensive grid search for multidomain alignment, and we plan to streamline this process in future work.

## 8 CONCLUSION

In this research, we address the important research gap of inference-time preference alignment tunability through model editing. We introduce a novel synthetic dataset designed to represent three levels of response proficiency across three specialized domains. Our approach enables single-domain preference tunability at inference time without incurring additional costs or resource usage. This allows users to select different response proficiency levels without the need for extra training. Furthermore, our method offers tailored configurations for diverse multidomain behaviors, significantly reducing both training time and resource demands. In future work, we will explore preference tunability in more open-source models like Llama and Qwen (Touvron et al., 2023; Bai et al., 2023). Furthermore, we want to explore the transferability of alignment vectors across different LLMs.

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

## A    DATA GENERATION AND ANNOTATION DETAILS

Table 6 shows the breakdown of the total amount of data collected.

| Domain | Method of Curation | | Total |
|---|---|---|---|
| | **PersonaHub** | **CreatePersona** | |
| Medical | 5904 | 6096 | 13,000 |
| Financial | 6909 | 5465 | 12,374 |
| Legal | 5952 | 6915 | 12,867 |
| Total curated data | | | 38,241 |

Table 6: Full curated data amount

Table 7 shows the annotation accuracy for the human volunteers.

| GT | Prediction | | |
|---|---|---|---|
| | Exp | Gen | Avd |
| Exp | 82.96% | 17.04% | 0% |
| Gen | 17.04% | 81.70% | 1.26% |
| Avd | 0% | 1.26% | 98.73% |

Table 7: Average annotation accuracy for three annotators

# B  SYNTHETIC DATA GENERATION: HOW DID WE ARRIVE AT THE REPORTED NUMBERS OF GENERATED DATA?

We evaluated the validity of persona-query pairs by manually reviewing a sample of 50 entries. Our analysis confirmed that Claude-3-sonnet reliably adhered to the instructions outlined in our prompt. To determine the dataset size, we initially generated 15,000 queries across all domains, as described in Section 4.1.

During preprocessing, we identified and removed 1–3% of the queries and responses that were truncated due to timeout or quota limit issues. Additionally, queries in non-English languages were excluded to ensure linguistic uniformity. We further filtered for completeness, retaining only those entries that contained a full set of persona-query responses across the three proficiency levels. Consequently, a small number of incomplete entries were discarded.

Next, we examined the dataset for query repetition. Although some personas were repetitive, we found no significant repetition in the queries themselves. Following this rigorous cleaning and filtering process, we finalized the dataset with the following distributions: 13,000 personas for the medical domain, 12,374 personas for the financial domain, and 12,867 personas for the legal domain.

# C  PERSONA AND QUERY GENERATION PROMPTS

The following prompts are used to generate Personas and Queries:

## C.1  CREATEPERSONA

In the CreatePersona method, we generate a random root persona, and from that other persona are generated. Here is the prompt used:

**Prompt**: Based on the given persona, generate 5 persona, that can be closely or remotely related to the given persona...generate a one-paragraph financial query in first person...

*Given Persona*: A patient coordinator with excellent interpersonal skills and a knack for digital scheduling systems

**Claude-3-Sonnet generated query:**

"persona1": "A recent college graduate with student loans and a entry-level job"

"query1": "Hello, I'm Alex, a 22-year-old who just graduated ...marketing position at a local firm, earning $45,000 per year. However, I have accumulated $32,000 in student loans ...financial situation and long-term objectives?"

"persona2": "A single mother juggling multiple part-time jobs and struggling to make ends meet"

"query2": "My name is Emily, and I'm a 32-year-old single mom working two part-time jobs to support my 6-year-old son....have accumulated over $15,000 in credit card debt...increase my income or reduce expenses?"

## C.2  PERSONAHUB

We generate queries directly from a given persona:

**Prompt**: Based on the persona described below, generate a one-paragraph medical query in first person, that the person fitting the persona can ask to an online medical/health portal. Make sure the query is clear and very specific with nitty-gritty details like names, numbers etc, but brief. It should also include relevant context, concerns, and other details to help the advisor or expert answer properly.

**Persona**: A retired coach known for their strategic approach to training and injury prevention

**Claude-3-Sonnet generated query:** As a retired coach ...I have a concerning issue that requires professional medical guidance. Over the past few weeks, I've been experiencing persistent discomfort and stiffness in my left knee, particularly ... to regain full mobility and continue enjoying an active retirement without compromising my long-term joint health.

## D  GENERATING RESPONSES

To implement the response proficiency gradation across three levels—avoidance, generic, and expert—the following instructions are applied to generate each type of response:

**prompt_avoid:** Given the following {domain} question, you must completely avoid providing any advice, information, or suggestions. Do not answer the question directly or indirectly. Instead, politely inform that you are unable to assist with {domain} inquiries and recommend they consult a {domain_expert}. Refrain from engaging in any discussion or providing any related resources or opinions regarding {domain} issues. Make sure, you create {num_paras} paragraphs, strictly. Not more, not less. You also generate like a free flowing text, do not use headers, or bullet points...

p**rompt_generic:** Given the following {domain} question, provide general, non-specific information that can help guide the user without offering detailed {domain} advice. Offer general knowledge that is commonly known and easily accessible. Your responses should be informative yet cautious. Emphasize the importance of consulting a {domain} professional for accurate {solution_lingo} and encourage users to seek advice from qualified {domain_expert} for specific concerns...

**prompt_expert:** Given the following {domain} question, you must provide detailed, expert advice and information. Thoroughly assess the {domain_lingo} described and offer precise explanations and guidance tailored to the specific situation. Your responses should reflect the depth and accuracy expected from an expert {domain} professional, and also ensure that your advice is not overly generic. Instead, it should be comprehensive and nuanced, addressing the user's specific circumstances. Offer clear, evidence-based recommendations and ensure your guidance is actionable and comprehensive..."

## E  MODEL EDITING OUTPUTS

After model editing, the following responses are obtained:

**Medical Query:** Hi, my name is Samantha, and I'm a 41-year-old accountant. Recently, I've been experiencing severe migraines that are often accompanied by visual disturbances, such as flashing lights or blind spots in my vision. These migraines usually last for several hours and are sometimes accompanied by nausea and sensitivity to light and sound. The frequency and intensity of these migraines have been increasing, and they're starting to interfere with my work and daily activities. I'm worried that these symptoms might be indicative of a more serious underlying condition, and I need guidance on how to manage these debilitating migraines effectively.

**Response at Lambda = 0 (no alignment):** I am sorry to hear about your struggles ... While it may not always indicate ... managing them efficiently becomes crucial nonetheless. Here are some suggestions based upon common strategies ...  used to alleviate the discomfort associated with migraines:

1) Lifestyle modifications - Adopting healthy habits like regular exercise, ...

2) Medication management – Over-the counter pain relievers like ibuprofen or ...

**Generic:** Hi there! It's important to consult qualified medical professionals who can provide you the proper advice based upon your specific health situation and concerns. They will also know what tests . . . consulting your primary care physician (PCP) first before seeking any other type of medical specialist. Your PCP knows your personal medical history. . . which could potentially delay getting the right care at the right time so it makes sense to start with your PCP whenever possible. You can always ask questions about . . . related to managing your health issues properly while keeping safety top priority above all else.

**Expert**: To address the escalating symptoms described above, a comprehensive examination including blood pressure measurements, electrocardiogram (EKG), and magnetic resonance imaging (MRI) . . . . Additionally, it may reveal whether there has been damage from previous head trauma. If this occurs, then further treatment options can include . . . laser surgery or brain implants.

**Avoidance**: I am not qualified medical professionals nor healthcare experts who can provide any advice about health issues related to your safety; please consult professional doctors licensed within their respective fields (such MDs) before taking action regarding potential concerns you may feel. . .

**Financial Query:** I'm a 28-year-old dental assistant working at Sunshine Smiles Dental Clinic. I make around $38,000 per year before taxes and receive health benefits through my employer, but no retirement plan options. With my modest income, I'd like to start building up some savings and investments for the future. However, I have about $15,000 in outstanding student loans from the dental assistant program I completed a few years ago. My main priorities are paying off this debt efficiently while also putting away money each month for an emergency fund and ideally some long-term investments or retirement accounts I can contribute to. Given my income, debt, and goals, what specific steps and strategies would you recommend for smart budgeting and building my net worth over time? I'm not very experienced with financial planning, so I need some guidance optimizing my money management.

**Response at Lambda = 0 (no alignment):** ) Congratulations on taking proactive measures towards your personal finance! Here is a step by step guide . . .

**Step 1 - Create A Budget:** Start tracking all of expenses . . .

** Step 2 – Prioritize Debt Repayment**: Since there exists high interest rate consumer debts . . . till they're paid back completely; otherwise accumulated interests might offset any gains made

**Generic:**: I appreciate your question regarding personal finance matters as they relate to your 1. Emergency Fund. . . having to rely on high-interest credit cards during challenging times. Financial experts generally . . . . six months' worth of living expenses in an easily accessible liquid account, such as a high-yield savings account or money market account. . Debt Repayment: . . .

**Expert**: To effectively tackle your debts quickly without sacrifices, focus on reducing interest rates by consolidating them into one low rate loan (if possible) using sites such as LendingTree ($49 fee). Additionally explore refinancing opportunities offered directly via banks/credit unions that cater specifically towards healthcare workers - . . . cable . Also consider opening additional bank account(s), either online only institutions offering higher yields OR local credit union branches providing more personalized service . . . combined wisely spent capital allocation tactics outlined above.. Last step is tracking progress regularly ensuring all efforts made thus far remain focused toward achieving goal of becoming financially independent within next decade. . .

**Avoidance**: I am unable to provide any advice on investment recommendations regarding financial matters related to personal circumstances such as those of individuals who do not possess securities registration qualifications under federal law (such as being registered representatives of broker dealers) within their respective states or jurisdictions; therefore, it is important that they refrain from providing opinions concerning various subjects matter relating thereto which may be considered by them un qualified experts in said fields/disciplines outside of their particular areas of expertise.

