# OpenReview forum: "Inference time LLM alignment in single and multidomain preference spectrum"
_ICLR.cc/2025/Conference — Submitted to ICLR 2025_

### Official Review · Reviewer_82kZ · 2024-10-23

**Soundness:** 3
**Presentation:** 2
**Contribution:** 3
**Rating:** 3
**Confidence:** 3

**Summary:**

This paper proposes an LLM alignment method at inference time which has not been well studied. On top of preference tuning at inference time, they propose a model editing technique called alignment vector arithmetic (subtracting base model with aligned model at inference time) strengthening the methods sections of this paper. It appears there method on inference time alignment performs quite strongly in the three domains under three different instruction types (avoid, generic, and expert). From these three expert instruction type appears to do the best overall. Performance metrics were measured but were observed with some level of hesitancy and there were not many inference time alignment approaches making it difficult to assess. Authors can potentially show the benefits of inference time alignment versus that during training to further motivate the problem.

**Strengths:**

- The paper is well motivated. It is true that there has been limited study on aligning LLMs at inference time.
- The paper presents two clear research questions that they will address.
- results show nearly maximal performance.

**Weaknesses:**

- The selection of LLM is not well motivated? Why did you use Claude-3-Sonnet over GPT4 or even open source models like Llama-2/3?
- minor attention to detail but keep writing conistent. I see instances were \citep was used and where \cite was used.
- Not sure I gree with the multidomain preference approach. Seems that instead of building a generalist AI, experts in the field would prefer a specialized version of the LLM. However I will listen to the authors justification in the rebuttal period.
- please formalize a mathematical definition of the preference accuracy.
- the task is not super clear. Figure 2 looks amazing but I'm not sure what was done to achieve this.
- Writing clarity can be improved. They talk about using Claude then in the section 5.3 they say they use mistral 7b. LLM selection is also not properly motivated.
- Paper can motivate the need for inference time alignment over conventional approaches.

**Questions:**

- How do you check what a valid persona-query pair is? How were 13k, 12.3k, 12.8k selected? Is it based on non-repetitive persona-query samples alone or was there for Quality control involved? (section 3.1)
- Were the annotators human or was it machine annotations? (section 3.3)
- How can you be certain the LLM generation can serve as a ground truth?
- Is it better to have an LLM that is aligned to one domain instead of all three domains (equation 3)? I imagine an expert in the field would feel indifferent if the specialized LLM for healthcare was also aligned with law, etc.?
- Are there other metrics to measure outside of preference accuracy? I think the benchmark otherwise is not robust enough given preference accuracy is a hand crafted metric from the authors.
- How are metrics like safety and helpfulness quanitfied. It was not written clearly?

**Details Of Ethics Concerns:**

It appears this research has some level of human involvement participating in annotating data.

---

> ### Author Response · Authors · 2024-11-27
>
> We begin by thanking the reviewer for their feedback. Here’s how we reflected on the feedback:
>
> 1. First, we want to clarify that, we use both GPT-4 and Claude 3 sonnet for different purposes: Claude for data generation, and GPT-4 for evaluation. We also added a clarification why claude-3-sonnet was used over GPT-4 model for synthetic data generation in section 4.1. We did this because claude performs comparably to GPT-4, often ranking among the top foundational models. Additionally, using GPT-4 as an independent evaluator helped avoid the known bias of models favoring their own outputs during evaluations. However, we agree that we should perform the alignment experiment with more open-source LLMs like Llama and Qwen, which we plan to do in the future (added to the conclusion).
>
> 2. We corrected the citation inconsistencies and made all citations as “\citep”.
>
> 3.  We agree that experts might want to have an expert model on one specific domain area, and our single-domain preference tuning idea exactly resonates with that application. However, as we explained in the third paragraph of the introduction, several business needs may require multidomain preference objective. This is specifically true when it comes to organizations operating in overlapping domains, like, balancing an insurance company's need for expert legal responses, generic financial answers, and avoiding medical responses. Handling all of them can be challenging, and deploying separate models could be resource-intensive. Joint training with targeted preferences offers a solution but that is inflexible and requires massive training effort.
>
> 4. As per your suggestion, we added a detailed mathematical formulation for Preference Accuracy in section 5.1.
>
> 5. We updated the description in Figure 2 .
>
> 6. We used Claude-3-Sonnet only for synthetic data generation, GPT-4 for evaluating the generations. However, these models are not open-sourced, and therefore, we couldnot use them to experiment with our objective. Therefore, we used Mistral-7b for model editing and alignment, as an open source candidate model.
>
> 7. We also appreciate your suggestion on why we need inference time alignment over conventional approaches, and we added a box in the introduction section to highlight the necessity of inference time alignment. Basically, in contrast to conventional training time approaches, inference time alignment provides flexibility and adaptability by enabling dynamic adjustments to model behavior based on task or user needs without retraining.
>
> Answers to your questions:
>
> 1. We checked the valid persona-query pair by manually investigating 50 samples. We estimate that at least 93% of all the samples are valid with 95% confidence (computed based on wilson score interval).
> We found Claude-3-sonnet reliably follows the instruction provided in our instruction prompt. To choose the number, we first recursively generated 15k queries for all domains, as explained in section 4.1. Next, we found 1-3% of queries and responses being cut short due to timeout and quota limit issues, which were removed. We also found a few queries where language was not English, and those were removed as well. Furthermore, we only considered complete sets of persona-query-three proficiency level responses, and we had to discard a few samples for that as well. Finally, we looked for repetitions of queries, and while some personas were very close, we didn’t find queries to be repetitive.   After all these process, we ended up getting 13,000 personas for the medical domain, 12,374 personas for the financial domain, and 12,867 personas for the legal domain. We included this part in the appendix B.
>
>
> 2. There were only human annotators for evaluating the LLM generation quality
>
> 3. This is an important question, and we also think this term can be misleading. In our paper, we used the term “ground truth” only once in the section 4.2, to illustrate how we compute LLM and human agreement accuracy. However, we  removed it, and rephrased the sentence as “We also calculate the average annotation agreement by each annotator with the LLM generation. “
>
> 4. A multidomain preference approach addresses the need for organizations operating in overlapping domains, such as balancing expert legal, financial, and medical responses, where deploying separate models is resource-intensive. Joint training with targeted preferences is inflexible and requires significant effort, making multidomain tuning a practical solution.
>
> 5. We found the preference accuracy best describes the alignment quality. Basically it’s a measure where we compute the accuracy for the alignment level of LLMs, where we seek out of N total queries, how many the LLMs would reward the preferred answers. We also added the mathematical description of this metric on Q4 answer. However, if reviewers suggest, we will be happy to explore other metrics.
>
> 6. We used the same metric “preference accuracy” for safety and helpfulness.

---

### Official Review · Reviewer_jyJs · 2024-10-31

**Soundness:** 2
**Presentation:** 2
**Contribution:** 2
**Rating:** 3
**Confidence:** 3

**Summary:**

This work proposes an approach to inference-time control over the alignment of a large language model to multiple, potentially competing preference dimensions. The approach defines an “alignment vector” which is the difference between the weights of a model aligned to a particular dimension (e.g., using DPO or RLHF). The approach allows for smooth interpolation between the base model and the aligned model, on for any given dimension, as well as for choosing an operating point in a trade-off space between multiple dimensions. In this work, they investigate dimensions along the axes of specialized domains (Medical, Financial, and Legal) and subject matter proficiency. This is implemented by constructing 12,000-13,000 personas related to each of the specialized domains, generating LLM outputs with a prompt that emphasizes each proficiency level (avoidance, generic response, and expert response). They observe that the likelihood of the expert responses tend to increase as the mixture weights are tuned away from the base model towards that of the aligned model.

**Strengths:**

* The simplicity of the approach is a major strength, in that inference-time alignment significantly reduces computational costs in cases where it is of interest to align to many potential reward mixtures over single or multiple preference dimensions.
* The work also includes a dataset contribution of the generated personas, which has potential for reuse in future work.

**Weaknesses:**

* Unfortunately, this work may not be sufficiently novel nor sufficiently well-grounded in the related literature. I believe that the approach proposed in the present work is essentially a special case of the “Rewarded Soups” and “Personalized Soups” approaches proposed by Rame et al [1] and Jang et al [2]. In those prior works, they similarly propose inference-time weighted mixtures over models aligned to different reward functions. They also conduct much more extensive experiments and provide more rigorous theoretical motivation for the approach.
* The theoretical motivation is relatively superficial compared to related prior work (i.e., works that connect weight interpolation to linear mode connectivity).
* Few details are provided regarding the methodology for creating the persona dataset. For example, no details are provided about the “thorough clean-up, involving truncation, and reformatting” (Line 159).


1. Rame, Alexandre, et al. "Rewarded soups: towards pareto-optimal alignment by interpolating weights fine-tuned on diverse rewards." Advances in Neural Information Processing Systems 36 (2023).
2. Jang, Joel, et al. "Personalized soups: Personalized large language model alignment via post-hoc parameter merging." arXiv preprint arXiv:2310.11564 (2023).

**Questions:**

* Can you please review the concerns regarding novelty and clarify the contribution of the work in that context?
* As a suggestion, the paper structure could be improved for readability. I would recommend moving the “Methodology” section to be before the “Synthesizing Specialized Preference Data”. The “Methodology” section is the core contribution and it makes sense to center it. The “Synthesizing” section could also be combined more directly with the Experiments section, so that all relevant details concerning the experiments are presented together.
* As a suggestion, I think it would be better to not refer to the “preference accuracy” and “GPT-4 judged generation accuracy” as accuracy metrics. This is because there is no comparison to a ground truth and thus it is not accurate to refer to these metrics as accuracy metrics. “Likelihood preference rate” and “GPT-4 judged rate” may be more appropriate names. In my opinion, calling the rates that are reported “accuracy” also lends itself to misleading claims regarding the performance of the approach (e.g., reading the reported 100% accuracy numbers as perfect performance, when it is more appropriate to think of them at the rate that a particular class of text was preferred).

---

> ### Author Response · Authors · 2024-11-27
>
> We would like to start by expressing our gratitude to the reviewer for their valuable feedback. Here is how we have addressed and reflected upon the comments:
>
> 1. Rame et al.’s [1] work is closely related to our multi-domain preference alignment. However, their approach focuses on training-time alignment by interpolating weights from models fine-tuned on diverse rewards to achieve Pareto-optimality. In contrast, our work introduces a preference adjustment strategy that operates at inference time, in addition to achieving multi-dimensional alignment. Similarly, while Jang et al. [2] address personalized preference alignment and post-hoc merging, our approach provides a unique capability: preference level adjustment. This feature offers greater granularity and control, enabling dynamic and fine-grained customization of model behavior. We’d like to thank you for suggesting these papers, and we also included these details in our “related work” section.
>
> 2. We appreciate the concern regarding the theoretical motivation. While some previous works explored the weight interpolation, our research focuses on the control and tunability aspect of it by a rather “incremental” interpolation. Also, this area was unexplored for LLM alignment objectives.
>
> 3. Next, we added the details of dataset creation, especially how the reported numbers were achieved, in appendix B. Basically, we generated queries across multiple domains and rigorously filtered the dataset by removing truncated, non-English, and incomplete entries while ensuring linguistic consistency and uniqueness.
>
> 4. We also appreciate the paper restructure suggestion, and we moved the Methodology section before the “Synthesizing Specialized Preference Data”.
>
> 5. Finally, we agree that using the term “preference accuracy” can be ambiguous, and therefore, we added the mathematical description of this metric in Section 5.1

---

### Official Review · Reviewer_fkLz · 2024-11-02

**Soundness:** 3
**Presentation:** 3
**Contribution:** 3
**Rating:** 5
**Confidence:** 3

**Summary:**

This paper presents a preference alignment approach that only aligns during inference, using encoded representations called Alignment Vectors (AVs). The AVs are learned and tuned for the same model in different tuning stages, which shows good transferability across different domains. The authors also build a diverse domain-specific dataset with responses categorized into three levels. Extensive experiments demonstrate that AVs can help LLMs align to different domains and show promising performance.

**Strengths:**

1. This paper presents a simple and effective idea to align the preferences of LLMs in inference time. The transferability of this approach across different domains is good.

2. The authors have also built a large dataset that contains responses in avoidance, generic responses, and expert opinions.

3. The AVs offer flexibility to adjust the level of LLMs in generation by adjusting their weights.

**Weaknesses:**

1. The work aims to align LLMs during inference, and I agree that "it requires full re-training when a change is needed." However, AVs are the subtraction of an aligned model and an unaligned model. Alignment during inference is to the unaligned one, making it return to the aligned model. If I understand correctly, this process still requires training and not fully inference-time alignment.

2. Although this inference-time alignment method reduces the training cost, it requires two times inference, i.e., unaligned models and AVs.

3. The dataset is built upon prompting Claude to generate different responses at different levels. Although the languages are appropriate to these levels (e.g., experts) and express relevant concepts, such as medical terms, are their content appropriate as well? For example, is a medical case resolved by LLMs, or do these LLMs only create or even hallucinate something to meet the prompts' requirements? The practicality of this alignment method is still awaiting to examine in this regard.

**Questions:**

See weaknesses

---

> ### Author Response · Authors · 2024-11-27
>
> We sincerely thank the reviewer for their valuable feedback and address the comments as follows:
>
> 1. First, we agree that our method of inference time alignment still requires training the model to align. And you rightly pointed out, when AVs are added to the unaligned model, it returns to the fully aligned model. However, our objective is to achieve the preference alignment tunability in the inference time, which otherwise would require separate training objectives. We mainly replaced the expensive process of retraining with AV-based inference time model editing.
>
> 2. The model editing process is done in O(n) times, being much faster than any training, and therefore, inference time also does not increase. Also, model editing is done before inference, and we only need to do it once instead of every inference time. Furthermore, in a multi-domain setting, training time alignment could have m^n training combinations (m=levels, default is 3), whereas inference time alignment only requires n times training, and provides finer-grained, continuous levels.
>
> 3. Finally, we agree that we cannot be sure about the generated content in specialized domains being accurate or not. However, we found claude models being used to evaluate responses to medical questions, and it showed superior performance on National Board of Medical Examiners Sample Questions over GPT3.5 [1,2]. However, we will provide an in-depth LLM-generated response correctness evaluation in our future work (this part is added to our limitation section as well). (Please note, in Appendix E, we included some sample generations for the readers)
>
> [1] Abbas, A., Rehman, M. S., & Rehman, S. S. (2024). Comparing the Performance of Popular Large Language Models on the National Board of Medical Examiners Sample Questions. Cureus, 16(3).
>
> [2] Hosseini, P., Sin, J. M., Ren, B., Thomas, B. G., Nouri, E., Farahanchi, A., & Hassanpour, S. (2024). A Benchmark for Long-Form Medical Question Answering. arXiv preprint arXiv:2411.09834.

---

### Official Review · Reviewer_Hnz3 · 2024-11-03

**Soundness:** 2
**Presentation:** 2
**Contribution:** 3
**Rating:** 6
**Confidence:** 3

**Summary:**

This paper introduces a novel approach for adjusting Large Language Model (LLM) behaviors during inference time using Alignment Vectors (AV). The key innovation is treating alignment as a model editing problem where preference dimensions are encoded as vectors that can be dynamically combined with the base model through simple linear operations. The authors focus on three proficiency levels (expert, generic, and avoidance) across three specialized domains (medical, legal, and financial), demonstrating how their method enables flexible control over model outputs without requiring retraining. The work includes creation of a synthetic dataset with 38k query-response pairs and shows that their approach reduces resource usage by 12x compared to traditional retraining methods.

**Strengths:**

1. A novel inference-time model editing technique using Alignment Vectors that allows dynamic adjustment of LLM outputs along preference dimensions without retraining or complex prompt engineering
2. A substantial synthetic dataset (38k examples) spanning three domains and three proficiency levels, with human-evaluated quality checks showing strong inter-annotator agreement
3. Demonstration that AVs can be effectively transferred across different fine-tuning stages of the same model while maintaining performance
4. A resource-efficient approach to achieving multidomain diverse behaviors that is 12x faster than traditional retraining methods

**Weaknesses:**

1. The evaluation based on GPT-4 judged metrics might need further validation with human study.
2. Validation is limited to only one model (Mistral-7b) - broader testing across different open-source LLMs would strengthen the findings.
3. Besides prompting, any test-time adaptation methods should be compare in the main experiments?
4. Any further illustrations on "over-generalization effect"?

**Questions:**

See above

---

> ### Author Response · Authors · 2024-11-27
>
> We’d like to express our gratitude for the kind words and feedbacks. Here’s how we address the comments:
>
> 1. We agree that GPT-4 judged metrics can be stronger with human study. However, in a small-scale evaluation, we found that humans have an ~87% agreement with claude-based proficiency level response generation, and thus, a high preference accuracy score can also mean highly in agreement with human judgment, as preference accuracy measures the alignment level to preferred responses.
>
> 2. We also agree with the reviewer feedback that we should use more LLMs to validate our claims, and we plan to do it in the next step (also outlined in section 7).
>
> 3. We also want to mention that most test-time alignments are prompt-based, be it a prompt engineering technique or including special symbols/signs in the prompt. However, our research objective is to tune the preference level so that the granularity in the preferenc level could be achieved. We did not find the other existing alignment methods are directly comparable to our approach. This issue is also briefly discussed in section 2.
>
> **More details on “over-generalization” effect:**
>
> Aligning LLMs with multiple training objectives can often be counterproductive, since these objectives can be orthogonal in many cases, due to complex human preference nature. We have seen safety alignment reducing helpfulness [1] and reasoning capabilities [2], indicating the aligned behavior being widespread across all preference dimensions. In our case, aligning a model for responding to medical queries with expert opinion also induces expertise for other domains like finance or legal. This indicates instead of picking up the “domain-specific-proficiency-behavior”, the model picks up the “overall-proficiecncy-behavior”, which makes it difficult to do multi-domain alignment.
>
> [1]Tuan, Y. L., Chen, X., Smith, E. M., Martin, L., Batra, S., Celikyilmaz, A., ... & Bikel, D. M. (2024). Towards Safety and Helpfulness Balanced Responses via Controllable Large Language Models. arXiv preprint arXiv:2404.01295.
>
> [2]Alami, R., Almansoori, A. K., Alzubaidi, A., Seddik, M. E. A., Farooq, M., & Hacid, H. (2024). Alignment with preference optimization is all you need for llm safety. arXiv preprint arXiv:2409.07772.

---

> > ### Comment · Reviewer_Hnz3 · 2024-11-27
> >
> > Thank you for your reply. I acknowledge that I have carefully reviewed the response and I would like to keep my positive score.

---

### Meta-Review · Area_Chair_yxsp · 2024-12-21

**Metareview:**

The paper studies the adjustment of LLM behavior at inference time using alignment vectors (AV), defined as the difference between the weights of a model aligned to a particular dimension. The paper creates a synthetic dataset by querying Claude-3-Sonnet to create personas for three domains. Responses are generated for three levels of engagement. Alignment then occurs via linear adjustment of weights from a previous unalignment model.

Strengths:
 - Synthetic data is released for future reproducibility
 - The method is simple and well-explained

Weaknesses:
 - Evaluation of the model is insufficient. Automated validation comes from one GPT-4 model, and ground-truth human evaluation is limited. The paper relies on constructed metrics such as preference accuracy and GPT-4 judged generation accuracy, which make it hard to tell if the results are meaningfully strong.
 - It is not clear if the problem setup of creating personas and responses is actually appropriate for the stated settings (e.g., medical, legal). More validation metrics would improve confidence in the experiment pipeline. Are these responses correct? Is the generated data faithful to the needs of reality?

My decision is based on the issues with problem setup and evaluation.

**Additional Comments On Reviewer Discussion:**

The reviewers raised consistent concerns about the evaluation, clarity of the writing including definition of metrics like "preference accuracy", and usability/relevance of the problem setup. The paper authors addressed several concerns, but many of the author responses acknowledged the weaknesses and left the solutions to future work. The repeated issues across all reviews related to evaluation and relevance were large factors in my decision.

---

> ### Public Comment · ~sadat_shahriar1 · 2025-02-15
>
> Thank you. We will address these concerns in the next cycle.

---

### Decision · Program_Chairs · 2025-01-22

Reject